# Combining Functional Genomics and Whole-Genome Sequencing to Detect Antibiotic Resistance Genes in Bacterial Strains Co-Occurring Simultaneously in a Brazilian Hospital

**DOI:** 10.3390/antibiotics10040419

**Published:** 2021-04-11

**Authors:** Tiago Cabral Borelli, Gabriel Lencioni Lovate, Ana Flavia Tonelli Scaranello, Lucas Ferreira Ribeiro, Livia Zaramela, Felipe Marcelo Pereira-dos-Santos, Rafael Silva-Rocha, María-Eugenia Guazzaroni

**Affiliations:** 1Department of Biology, Faculdade de Filosofia, Ciências e Letras de Ribeirão Preto, Universidade de São Paulo, Av. Bandeirantes 3900, Ribeirão Preto, SP 14049-901, Brazil; tiago.borelli@usp.br (T.C.B.); gabriel.lencioni.lovate@uni-jena.de (G.L.L.); anatonelli@usp.br (A.F.T.S.); lucasfribeiro@usp.br (L.F.R.); 2Department of Pediatrics, University of California San Diego, San Diego, CA 92161, USA; lzaramela@ucsd.edu; 3Department of Cell and Molecular Biology, Faculdade de Medicina de Ribeirão Preto, University of São Paulo, Av. Bandeirantes 3900, Ribeirão Preto, SP 14049-900, Brazil; felipemarcelo@usp.br (F.M.P.-d.-S.); silvarochar@usp.br (R.S.-R.)

**Keywords:** whole-genome analysis, functional genomics, resistome, mobilome, antibiotic resistance genes

## Abstract

(1) Background: The rise of multi-antibiotic resistant bacteria represents an emergent threat to human health. Here, we investigate antibiotic resistance mechanisms in bacteria of several species isolated from an intensive care unit in Brazil. (2) Methods: We used whole-genome analysis to identify antibiotic resistance genes (ARGs) and plasmids in 34 strains of Gram-negative and Gram-positive bacteria, providing the first genomic description of *Morganella morganii* and *Ralstonia mannitolilytica* clinical isolates from South America. (3) Results: We identified a high abundance of beta-lactamase genes in resistant organisms, including seven extended-spectrum beta-lactamases (OXA-1, OXA-10, CTX-M-1, KPC, TEM, HYDRO, BLP) shared between organisms from different species. Additionally, we identified several ARG-carrying plasmids indicating the potential for a fast transmission of resistance mechanism between bacterial strains. Furthermore, we uncovered two pairs of (near) identical plasmids exhibiting multi-drug resistance. Finally, since many highly resistant strains carry several different ARGs, we used functional genomics to investigate which of them were indeed functional. In this sense, for three bacterial strains (*Escherichia coli*, *Klebsiella pneumoniae*, and *M. morganii*), we identified six beta-lactamase genes out of 15 predicted in silico as those mainly responsible for the resistance mechanisms observed, corroborating the existence of redundant resistance mechanisms in these organisms. (4) Conclusions: Systematic studies similar to the one presented here should help to prevent outbreaks of novel multidrug-resistant bacteria in healthcare facilities.

## 1. Introduction

Microbial resistance to antibiotics is a growing global concern. Established protocols in clinics to fight nosocomial infections include the isolation of microorganisms from patient samples to allow their identification and to determine antibiotic susceptibility [1]. However, this process is time-consuming (taking 48 h or more), and prone to pathogen misidentification [2]. Therefore, the advent of next-generation sequencing (NGS) tools has allowed the rise of novel approaches to identify microbial pathogens and to fight infection [3]. Thus, in the last two decades, whole-genome sequencing (WGS) of microbial pathogens has moved from being used mainly as a basic research tool to understand the biology and evolution of pathogens [4,5], to a valuable tool to investigate outbreaks in hospitals and nosocomial infection pathways [6,7,8,9]. For diagnostics purposes, current WGS technologies can be cost-effective even for slow-growing pathogens such as *Mycobacterium tuberculosis*, providing faster and more accurate results even for antibiotic resistance determination [10]. Additionally, culture-independent methods based on clinical metagenomics can be used to identify several pathogens from nucleic acids extracted from patient samples without the need for microbial isolation [11,12,13]. Furthermore, recent progress in the use of artificial intelligence tools has allowed for the construction of computational models that can predict with high accuracy the antimicrobial susceptibility of microbial pathogens based on WGS data [14,15,16].

While routine WGS analysis for microbial identification is not a worldwide reality, it has been extensively used to investigate microorganism population structures on different scales. For example, Arias and coworkers, using WGS analysis from 96 methicillin-resistant *Staphylococcus aureus* (MRSA) samples from nine countries in Latin America, demonstrated a high degree of variation in the genome of different isolates from those countries [17]. The same study indicated that among sampled hospitals, those in Brazil presented a higher incidence of MRSA strains (up to 62%). Similarly, recent work by David and coworkers investigated the pathway of the nosocomial spread of *Klebsiella pneumoniae* in 244 hospitals in 32 European countries [18]. Using a well-defined sampling strategy and WGS analysis of more than 1700 *K. pneumoniae* strains, the authors could quantify the role of intra-hospital pathogen dissemination, as well as some potential pathways for the introduction of novel strains from the United States to Europe. In addition to those examples, large-scale WGS analysis has been used to investigate the molecular adaptation to different hosts, as in the work by Arimizu et al., where the authors analyzed *Escherichia coli* strains from human versus bovine samples [19].

Another key process playing a significant role in the rise of new microbial threats is the propagation of mobile virulence and antimicrobial resistance factors within these populations mediated especially by plasmids and transposons [5,20]. Therefore, the rapid evolution of plasmids through structural rearrangements, acquisition of virulence genes, plasmid fusions, and propagation to pathogens can account for the fast dissemination of super-virulent or super-resistant bacteria [21,22,23]. Understanding the very dynamic and complex processes could hold the potential for the design of new drugs aiming at reducing plasmid propagation between pathogens [24]. While the use of WGS is currently growing worldwide, most studies (especially in Brazil) have been restricted to some particular species (such as *K. pneumoniae* or *S. aureus*), without considering the importance of other species as repositories of antibiotic resistance genes (ARGs) in hospital settings. In our study, we collected clinical bacterial strains to search for ARGs and their association with mobile genetic elements. We identified ARGs with high spreading potential by combining whole-genome sequencing and functional genomics. Here, we investigate 34 bacterial strains at the genomic level from 18 different species (and 11 genera) isolated over the same two weeks at a public reference hospital in Brazil. WGS analysis indicated that many strains are only distantly related to those available in public databases. We aimed to identify ARGs and plasmids harboring these elements, as well as evidence for common resistance mechanisms shared between strains from the same or different species. We were able to identify several ARG-harboring plasmids, two of which were present in both Gram-negative and Gram-positive strains, and seven beta-lactamases located in multiple hosts with 100% identity at the nucleotide level, two of which were inferred as being active using functional genomic library screening. In addition, comparative sequence analysis identified a novel IncFII *K. pneumoniae* plasmid harboring two ARGs, potentially indicating a recent introduction from Asia to Brazil.

## 2. Materials and Methods

### 2.1. Sample Collection

Samples were taken at the Ribeirão Preto Clinics Hospital (HCRP, Ribeirão Preto, Brazil), a tertiary reference hospital in Latin America with 920 beds and 34,000 hospital admissions per year. Samples were isolated from different patients hospitalized in weeks 44 and 48 of 2018. Strains were obtained from different samples, as indicated in Figure 1A. Thirty-five strains were randomly selected from a total of the 105 strains that were available and represent different species, with a major prevalence of the *Klebsiella* genus, *E. coli*, *Pseudomonas aeruginosa,* and the genus *Staphylococcus*. After strain characterization by Vitek 2, samples were inactivated and used for genomic DNA extraction and sequencing, as indicated below.

### 2.2. DNA Extraction and Genome Sequencing

Total genomic DNA was extracted using Wizard Genomic DNA Purification Kit (Promega, Madison, WI, USA) following the manufacturer’s instructions. Figure 1B schematically represents the overall strategy used for WGS analysis. The DNA concentrations were measured fluorometrically (Qubit^®^ 3.0, kit Qubit^®^ dsDNA Broad Range Assay Kit, Life Technologies, Carlsbad, CA, USA). Purified DNA from 34 isolates was prepared for sequencing using the Nextera XT DNA Library Prep Kit (Illumina, San Diego, CA, USA). Libraries were assessed for quality using the 2100 Bioanalyzer (Agilent Genomics, Santa Clara, CA, USA) and subsequently sequenced using HiSeq 2 × 150 bp cycle kits (Illumina, San Diego, CA, USA). On average, 5.5 million reads were generated per library. Adapters were trimmed using Trimmomatic v0.36. Samples were filtered for possible human contamination by aligning the trimmed reads against reference databases using Bowtie2 v2-2.2.3 with the following parameters: (-D 20 -R 3 -N 1 -L 20 very-sensitive-local). Overlapped reads were merged using Flash version 1.2.11. Merged and unmerged reads were assembled using Spades v3.12.0 with the following parameters: (-k 21, 33, 55, 77, 99, 127 --merge). Genome quality (completeness and contamination) was evaluated using CheckM v1.0.7 and QUAST. Genome annotations were performed using Prokka v1.11 with default parameters. Amino acid sequences of all genes identified using Prokka were aligned to the National Database of Antibiotic Resistant Organisms (NDARO) database obtained from NCBI (March 2020). The alignment was performed using Diamond v0.8.24 with the following parameters: (blastx -k 5 -f 6 –E value 0.001). Alignments with ≥60 similarity score were selected for further analysis. Quality assessment of sequenced genomes is provided in Appendix A. All genomes are available at the NCBI under the BioProject number PRJNA641571.

### 2.3. Identification of Antibiotic Resistance Genes, Plasmids, and Phylogenomic Analysis

The identification of antibiotic resistance genes was performed using the ABRicate pipeline by searching annotated genes using reference databases (ARG-ANNOT, NCBI AMRFinderPlus (https://www.ncbi.nlm.nih.gov/bioproject/PRJNA313047 accessed on 1 December 2020), CARD, and ResFidner), as well as DeepARG [25]. The identification of plasmids was performed using Plasmidfinder, and contigs harboring ARGs and plasmid-related genes were further analyzed in detail. For genomic comparison, reference and assembly genomic data were downloaded from the NCBI databank. Phylogenomic analyses were performed using Parsnp and Gingr [26] and phylogenetic trees were visualized using iTOL [27].

### 2.4. Genomic Library Construction

For the cloning of the genomic DNAs in the pSEVA232 [28,29,30] vector, 2 µg of genomic DNA from each strain were digested with Sau3AI. Meanwhile, pSEVA232 digestion using BamHI and further dephosphorylation were performed. Genomic fragments from 1.5 to 6 kb and the linearized pSEVA232 vector were selected and incubated with the T4 DNA Ligase enzyme in a 2:1 insert/vector ratio. Then, ligations were transformed in the electrocompetent *E. coli* DH10B [31] with a MicroPulser electroporator (Bio-Rad, Hercules, CA, USA). The resulting libraries were analyzed for the percentage of plasmids carrying genomic DNA and the average size of the insert they contained.

### 2.5. Determination of Minimum Inhibitory Concentrations (MICs)

The MICs were determined in the same culture medium (solid LB) and conditions of the screenings, employing serial dilution of the test antibiotics (amoxicillin, oxacillin, or penicillin G). Solid medium plates supplemented with kanamycin (50 µg mL^−1^), IPTG (100 μM), and dilutions of each beta-lactam antibiotic were inoculated with approximately 2.5 × 10^6^ colony forming units (CFUs) of *E. coli* DH10B harboring the pSEVA232 vector. We performed dilutions of antibiotics and culture media according to the CLSI M100 supplement [32].

### 2.6. Screening and Phenotype Confirmation

Three pools of clones (2.5 × 10^6^ clones per plate) were plated from each library on solid LB supplemented with kanamycin (50 µg µL^−1^), IPTG (100 µM), and inhibitory concentrations of each beta-lactam antibiotic (8 µg mL^−1^ for amoxicillin, 32 µg mL^−1^ for penicillin G, or 256 µg mL^−1^ for oxacillin). Positive clones were cultured in liquid LB medium supplemented with kanamycin (50 µg mL^−1^) for extraction of plasmid DNA with the Wizard Plus SV Minipreps DNA Purification System Kit (Promega, Madison, WI, USA). Plasmids with unique EcoRI/HindIII restriction patterns were subjected to re-transformation and phenotypic confirmation by streaking the clones in solid LB medium supplemented with kanamycin (50 µg mL^−1^), IPTG (100 µM), and the test antibiotics.

### 2.7. Extraction of Insert Sequence from Assembled Genomes

Plasmid DNA was sequenced on both strands by primer walking using the ABI PRISMDye Terminator Cycle Sequencing Ready Reaction kit (PerkinElmer, Waltham, MA, USA) and an ABI PRISM 377 sequencer (Perkin-Elmer, Waltham, MA, USA) according to the manufacturer’s instructions. A python script was developed to extract and annotate inserts from Sanger reads, which is available on GitHub (https://github.com/tiagocabralborelli/Borelli-et-al-insert-finder accessed on 1 December 2020). First, the algorithm converted ab1 files to the fasta format and searched for the read sequence in the assembled genome using BLAST+ [33]. We then extracted the inter-read region to a fasta file, which was then annotated by PROKKA [34].

## 3. Results and Discussion

### 3.1. WGS Analysis of Clinical Strains Isolated Over the Same Two Weeks

We selected 34 bacterial strains isolated from different patient samples and performed WGS as represented in Figure 1. While well-studied pathogens such as *K. pneumoniae*, *E. coli*, *P. aeruginosa,* and *S. aureus* were well represented in the sampling, we were able to analyze pathogens with very little representative genomic information in January 2021. For example, two *Ralstonia mannitolilytica* strains were sequenced, but only nine complete or draft genomes were available at the NCBI. Other relatively underrepresented strains were *Streptococcus gallolyticus* (28 genomes available) and *Morganella morganii* (96 genomes). These numbers contrast with those of *K. pneumoniae*, *E. coli*, or *S. aureus*, where 8000–20,000 genome sequences are available. This evidence indicates that many clinically relevant pathogens have been underrepresented in WGS analysis efforts worldwide. For instance, phylogenetic analysis of *Burkholderia cepacia* 540A against all 166 available genomes in NCBI (Appendix A) indicates this strain is also very divergent from other strains but belongs to a branch formed by strains isolated from a patient with cystic fibrosis in the United Kingdom and endophytic bacteria isolated from Australia [35]. Therefore, some of the new genomes generated here demonstrate a significant diversity of some underrepresented microorganisms and should serve as reference sequences for future studies on clinical isolates in Brazil and South America.

### 3.2. Identification of Resistance Genes in Clinical Strains

We next analyzed the existence of ARGs in the sequenced genomes. Analysis of ARGs using the ARG-ANNOT database indicated a higher prevalence of beta-lactamase coding genes, followed by amino-glycosidases (Figure 2A), while analysis with the DeepARG tool [25] showed the multidrug category to be the most abundant on the genomes analyzed (Appendix A). We next analyzed the ARG distribution in the two most abundant groups, *E. coli* and *Klebsiella*. In the first case, each of the six *E. coli* strains displayed a unique set of ARGs from different categories, and five particular beta-lactamases (*bla*_TEM-105_, *bla*_OXA-1_, *bla*_KPC-2_, *bla*_CTX-M-15_, and *bla*_CMY-111_) were found in at least one genome (Figure 2B). Only one strain (*E. coli* 456A) did not present any of these five *bla* genes, and this strain was sensitive to all antibiotics tested by Vitek 2 (Figure 2B and Figure 3). Next, we investigated which beta-lactamase genes were conserved in the strains analyzed. For this, we compared all of the ~125 *bla* genes identified in Figure 2 and searched for those coding proteins with 100% identity at the amino acid sequence (Figure 3). Using this approach, we identified seven *bla* genes (*bla*_OXA-1_, *bla*_OXA-10_, *bla*_CTX-M-1_, *bla*_KPC_, *bla*_TEM_, *bla*_HYDRO_, and *bla*_BLP_) which have 100% aa identity between two or more strains. Notably, *bla*_OXA-1_, *bla*_CTX-M-1_, *bla*_KPC_, and *bla*_TEM_ were present in three to six strains from different species. Interestingly, both *bla*_OXA-1_ and *bla*_CTX-M-1_ were found in *E. coli*, *K. pneumoniae*, and *M. morganii*, which could reveal horizontal gene transfer between these strains in the past. From these seven genes, two (*bla*_OXA-1_ and *bla*_KPC_) were found to be active in the functional screening presented below. The ARG patterns identified in *E. coli* and *Klebsiella* are quite common and are often associated with widespread mobile genetic elements (MGE) with different levels of conservation, as seen in our work (Figure 3) [36,37,38,39].

### 3.3. Intrinsic Resistome and Hospital-Associated Resistome

Classically, antimicrobial resistance has been seen from a hospital-associated resistome point of view which comprises genes originated by selective pression due to antibiotic overuse and/or acquired from horizontal gene transfer (HGT). Their hosts are capable of thriving under treatment with antimicrobial substances above a clinically established concentration threshold (MIC). However, recent works found that some genes from different ontologies and not directly involved in antimicrobial response can provide resistance (the so-called intrinsic resistome) in environments apart from hospitals [40]. For instance, ABC-system efflux pumps extrude a myriad of harmful substances from within the bacterial cell and, by doing so, they reduce the inner antibiotic concentration [40,41]. Since the identification of the intrinsic resistome, new definitions such as the ecological threshold (ECOFF) arrived to classify whether a species is intrinsically resistant. In this work we chose ABRicate to search for ARGs since this tool only identifies acquired ARGs, therefore excluding the intrinsic resistome. Furthermore, resistance phenotypic testing in hospital conditions matches with the ARGs found using the ARG-ANNOT database. A similar strategy was applied by Zankari and colleagues [42], who were responsible for the ResFinder database. Furthermore, according to the Beta-Lactamase DataBase (BLDB) [43], only two beta-lactamases detected in this study belong to the intrinsic resistome of their hosts (Appendix A) (*bla*_OXY2-1_ and *bla*_SHV-187_ from *K. oxyteca* and *K. pneumoniae*, respectively). Finally, further experimental validations confirmed the increased MIC in the presence of the ARGs that we found in heterologous hosts, as discussed below.

### 3.4. Identification of ARGs Located in Plasmids

We next aimed to identify ARGs with potential mobilization through plasmids in the analyzed species. For this, we crossed the data from ARG-ANNOT with the prediction of plasmid elements generated by PlasmidFinder. Using this approach, we were able to identify nine potential plasmids from seven species associated with at least one ARG. As shown in Figure 4A, a ~42 kb plasmid (pKP98M3N42) harboring a *bla*_KPC-2_ and a *sat-2A* resistance determinant were identified in *K. pneumoniae* 98M3, and this plasmid also carries transposases, recombinases, and type IV secretion system genes. An identical plasmid (pKP125M3N44; 100% nucleotide sequence identity) was also found in *K. pneumoniae* 125M3 (Figure 4A and Appendix A). As mentioned before, this *bla*_KPC-2_ gene was identical in these two *K. pneumoniae* strains and in *E. coli* 126M3, but it was not possible to locate the ARG in a plasmid in the latter case. This might be due to limitations in our draft assemblies. In two recent studies, the authors demonstrated ARG-carrying plasmid transfer in clinical settings between bacteria of the same or different species, emphasizing the significance of revealing potential plasmid-mediated outbreaks to efficiently track horizontal ARG transmission in hospitals [44,45]. Once phages could transfer ARGs among their hosts we also employed Phaster [46], a web-service for phage identification, to search for the co-occurrence of phages and ARGs, but ARGS were found in phage-related regions of the genomes (Appendix A). A comprehensive HCRP phagesphere analyses would provide more information on this topic.

Another strain, *K. pneumoniae* 508B, harbors two plasmids with ~136 kb (pKP508BN15) and ~62 kb (pKP508BN34). Both plasmids harbor transposon elements, with the larger one harboring a *sul2* resistance gene and the smaller one with two ARGs (*qnr-S1* and *bla*_LAP__-2_, Figure 4B,C). In general, the coexistence of ARGs with transposon elements was also observed for two plasmids identified in *E. coli* strains (Appendix A) and *Staphylococcus warneri* 732B (Appendix A). Finally, two almost identical small plasmids (~2.3 kb; 99.94% nucleotide identity) were identified in *S. warneri* 732B and *Staphylococcus epidermidis* 452B, which harbor *aadC* and *ermC* resistance determinants (Appendix A). Taken together, these data demonstrate the potential for dissemination of many ARGs genes identified here, and the existence of identical or near-identical plasmids between different species could indicate that these elements have been mobilizing among some of these species. We then confirmed the presence of IncC and IncX3 replicons in the plasmids of *K. pneumonia* 508B and 98M3, respectively, with bacWGSTdb [47,48] (Appendix A), but no new replicons were found in association with antibiotic resistance genes.

### 3.5. Experimental Validation of Functional Beta-Lactamases from High-Resistance Bacteria

Once we distinguished several ARGs in the genomes analyzed, we decided to perform a functional screening to potentially discover unknown ARGs in the strains while also identifying which of these genes could confer resistance to a heterologous host. For this, we selected three strains (*E. coli* 126M3, *K. pneumoniae* 508B, and *M. morganii* 538A) to construct genomic libraries into laboratory *E. coli* DH10B (Table 1). The libraries were constructed into the broad host range vector pSEVA232, which contains a kanamycin resistance marker, a broad host range *oriV* with a medium-copy number, and a variant of the *lacZα* multiple cloning site with a *Plac* promoter [28,29,30]. In this way, the libraries generated during this study or individual plasmids of interest can be transferred to other bacterial strains for another screening or functional evaluation (Figure 5A). The use of a medium copy-number plasmid allows a closer assessment of the natural genetic context of ARGs, in contrast to other studies that use high-copy plasmids [50]. Accordingly, plasmids with lower copy-number and monomeric states also tend to be more stably inherited throughout bacterial populations [51]. We screened ~750,000 clones of each library against each antibiotic (amoxicillin, oxacillin, and penicillin G) and obtained 44 clones with unique sequences containing ARGs (Figure 5B and Table 2).

Clones containing the *bla*_KPC-2_ were by far more abundant in the screening (Table 2 and Figure 6). The genomic context of identified *bla*_KPC-2_ indicates that it is prone to horizontal transfer once it is flanked by transposases. Martínez et al. [52] described a framework to prioritize the risk of ARGs, the Resistance Readiness Condition (RESCon). The RESCon algorithm considers the similarity of an ARG to known genes, functional evaluation, the clinical relevance of the antibiotic, the presence of a mobile genetic element, and presence in a human pathogen. The characteristics of this ARG would categorize it as RESCon 1, an ARG with the highest possibility of thriving in a clinical setting [52]. Although a framework better describing the impacts of gene transfer to prioritize risk is needed [53], the RESCon classification indicates the clinical relevance of this ARG. With ARGs found in all three functional screenings, we were able to identify with high frequency six different *bla* genes using the functional approach presented here, with at least two of them being present in plasmids—*bla*_KPC-2_ and *bla*_LAP__-2_ (see next section (Figure 6)).

We propose that functional genomics can be combined with current approaches based on large-scale sequencing in order to better understand the functional aspects of ARGs. Recent developments in machine learning [25] have provided tools to find ARGs that can be used to detect ARGs that would not be otherwise identified with sequence similarity tools. Still, the databases used to train those tools are biased for specific antibiotic resistance classes such as beta-lactam, bacitracin, macrolide–lincosamide–streptogramin (MLS), and efflux pumps. Indeed, we could only find just a fraction of ARGs annotated through bioinformatics tools with our functional approach. This paradox could be due to the other ARGs not being functional in the experimental conditions used here or because our screening was not exhaustive enough to cover those sequences.

### 3.6. Mapping of Plasmids in the Brazilian Territory

To evaluate the biogeographic distribution of functional ARGSs from *K. pneumoniae*, we performed genomic comparative analysis including 50 plasmids deposited in the NCBI data bank and two of the plasmids identified in this study (pKP98M3N42, pKP508BN34). Distance tree analysis showed the presence of two distinguishable groups. Plasmid pKP98M3N42 (identical to plasmid pKP125M3N44, also identified in this work) is located in the first group (comprising seven strains, all reported in Brazilian cities from 2009 to 2015 except for one from the United States), which indicates that pKP98M3N42 should share structural features with the other plasmids positioned in this group (Figure 7A). Additionally, pKP98M3N42 shares 99.9–100% sequence identity with IncX3 plasmids of these seven strains, which have previously been shown to play an important function in mediating the horizontal transmission of *bla*_KPC-2_ genes among hospital-associated members of the Enterobacteriaceae family [54,55]. Moreover, IncX3 plasmids carrying *bla*_KPC-2_ genes were also reported in countries that are very distant geographically, such as the United States [56], Brazil [57,58], Australia [59], Italy [60], France [61], South Korea [62,63], China [64], and Israel and Greece [54] (to cite a few). Interestingly, although IncX3 self-transmissible plasmids are widespread globally, the evident diversification of plasmids in the first separated branch of the tree (Figure 7A) could indicate that plasmids from strains isolated in clinically relevant bacteria in Brazilian ground undergo their own structural reorganization. The group formed with *K. pneumonia* 98M3 is mostly related to isolates previously found in Brazil, as well as one strain from Japan and another from Thailand (Appendix A), which means that both the host and its derivative plasmid are established in Brazil.

On the other hand, distance tree representation of the 50 close plasmid sequences related to pKP508BN34 did not show the presence of evident different groups. BLAST analysis showed that plasmid pKP508BN34 is widely distributed in *K. pneumoniae* strains, with 99.9–100% sequence identity and query coverage ranging from 84% to 100% to the 18 closest related plasmids, with some associated to hypervirulent strains, such as strain KP58 bearing plasmid pKP58-3 (Figure 7B). The plasmid pKP508BN34, which is 63.067 bp in size and harbors various mobile elements that contain antimicrobial resistance genes including *qnrS1* and *bla*_LAP__-2_, belongs to the IncFII plasmid group. As seen in the branch highlighted in Figure 7B, most of the strains carrying the closest related plasmids to pKP508BN34 were described in different cities of China, as well as one in Germany, one in Japan, and three in Thailand, and were associated with diverse host diseases such as pneumoniae, pulmonary infection, urinary tract infection, intestinal infection, and diarrhea, according to the data deposited in the bioprojects of the NCBI. In some cases, related strains were reported in asymptomatic patients (strains TH164 carrying plasmid pTH164-3 and strain TH114 bearing plasmid pTH114-3, both from Thailand), which is reasonable since *K. pneumoniae* is also a member of the gut microbiota [65]. Mortality due to infection of *K. pneumoniae* has been very rare in the past. However, this pathogen’s fast evolution due to the gaining of hypervirulence plasmids has allowed this bacterium to cause severe community-transmitted infections in relatively young and healthy hosts since the late 1980s (54, 55). To the best of our knowledge, this is the first time that this plasmid has been reported in Brazil. Remarkably, in contrast to *K. pneumoniae* 98M3, the group formed with *K. pneumonia* 508B3 is mostly related to isolates previously found in places outside Brazil such as the United Kingdom, Austria, and the Czech Republic (Appendix A), which contrasts with the origin of the plasmids. Those findings show that plasmid and host came from different regions on the globe, suggesting acquirement of KP508BN3 by *K. pneumoniae* 508B.

As shown in Figure 8A,B, plasmids pKP98M3N42 and pKP508BN34 are highly structurally conserved between *K. pneumoniae* strains available in the databank. Plasmids pKP98M3N42 and pKP1253N44 (Figure 8A) are most similar to plasmids found in *K. pneumoniae* and *E. coli,* which have been frequently reported in Brazil (as in Figure 7A). Plasmid pKP508BN34 (Figure 8B) was most similar to plasmids only isolated from *K. pneumonia,* mostly identified in Asia (shown in Figure 7B) with a high degree of conservation. However, plasmid pKP508BN15 (which is present in *K. pneumoniae* 508B) presented a strong structural diversification in the region close to the antibiotic markers, and these changes seem to be related to the activity of the ISSpu21 transposon element located in this region (Figure 8C). Plasmid pKP508BN15 (Figure 8C) was most similar to plasmids found in several different bacterial species (*K. pneumoniae*, *Photobacterium damselae subsp. piscicida*, *Vibrio alginolyticus,* and *Vibrio cholerae*) reported in Asia, Africa, and North America. Both *sul2* and rRNA adenine n-6-methyltransferase (*ermC*) resistance markers are located in a highly divergent region of the plasmid and are not present in most related plasmids. Interestingly, most related sequences available in the database are from worldwide strains, including some isolated from Asia, Africa, North America, and Europe, but no sequences were found from South America.

## 4. Conclusions

Here, we sampled clinical bacterial strains to investigate the existence of ARGs and their association with mobile genetic elements. We identified several ARGs shared between strains from different species, and some of these ARGs were associated with large plasmids, mostly endowed with transposable elements. Instead of focusing on strains from the same species or genus, our approach considered strains co-occurring simultaneously into a hospital setup, aiming to identify circulating resistance mechanisms that could have been mobilizing in this environment. While our analysis does not provide unequivocal evidence that these resistance mechanisms are being mobilized among the analyzed strains, we found strongly conserved ARGs located in plasmids and associated with transposon elements that could represent potential mechanisms for the dissemination of antibiotic resistance among clinical strains.

Additionally, by using functional genomics, it was possible to investigate which of the ARG candidates identified in silico could be expressed in *E. coli* and associated with the resistance to beta-lactam antibiotics in a different genomic context. Furthermore, for many of the bacterial species analyzed here, we found a low number of available complete genome sequences in the NCBI database. Therefore, while 1000 genome sequences are available for classical pathogens (*E. coli*, *K. pneumoniae*, *P. aeruginosa*, etc.), other clinically relevant pathogens such as *M. morganii* and *B. cepacia* are underrepresented, which makes it challenging to track genomic events associated with the acquisition of pathogenicity elements or resistance mechanisms in hospital-associated infections. Finally, an analysis of plasmids from *Klebsiella* strains allowed for the identification of both well-known circulating variants in Brazil as well as new variants that seem to have recently been acquired from Asia. Thus, we argue that more systematic efforts should be made to monitor the introduction and propagation of mobile genetic elements harboring ARGs, especially in South America, in order to inform and guide policies to minimize and prevent outbreaks and respond properly.

## Figures and Tables

**Figure 1 antibiotics-10-00419-f001:**
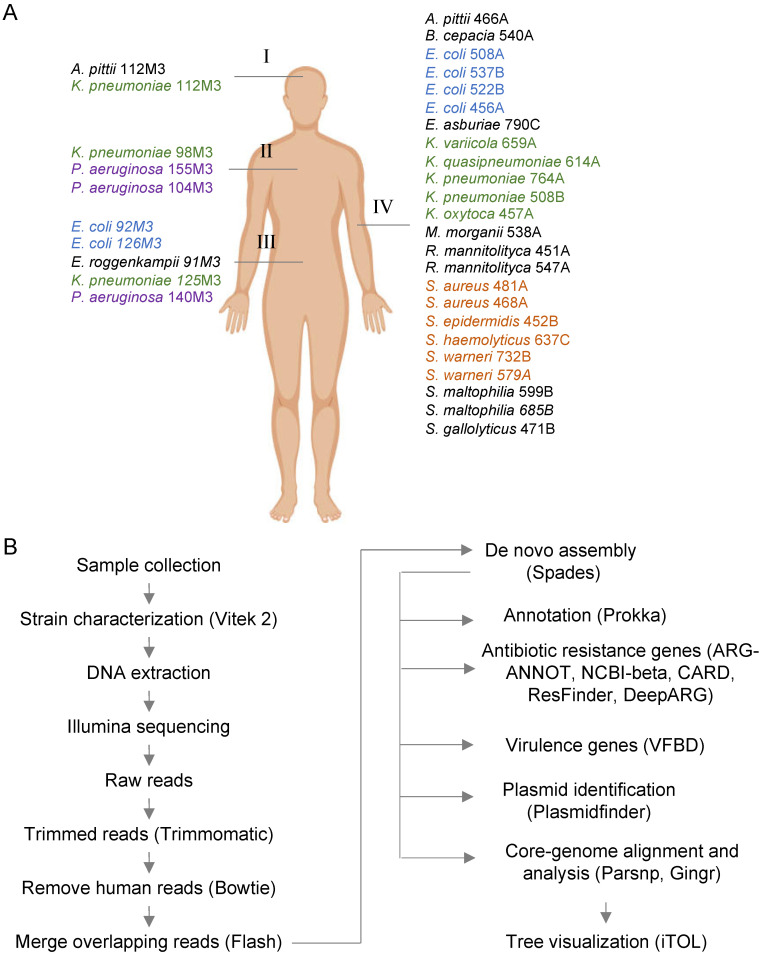
Overall strategy used for the whole-genome analysis of clinical strains. (**A**) In total, 34 bacterial strains were isolated from several samples, such as cerebrospinal fluid (I, 2 strains), bronchoalveolar lavage (II, 3 strains), ascitic fluid (III, 5 strains), and blood (IV, 25 strains). The four most common bacterial groups (*Klebsiella pneumoniae*, *Escherichia coli*, *Pseudomonas aeruginosa,* and *Staphylococcus* spp.) are colored. (**B**) Schematic representation of the main bioinformatic pipeline used for genome sequencing, assembly, annotation, and identification of antibiotic resistance genes (ARGs) and virulence factor and genomic analysis.

**Figure 2 antibiotics-10-00419-f002:**
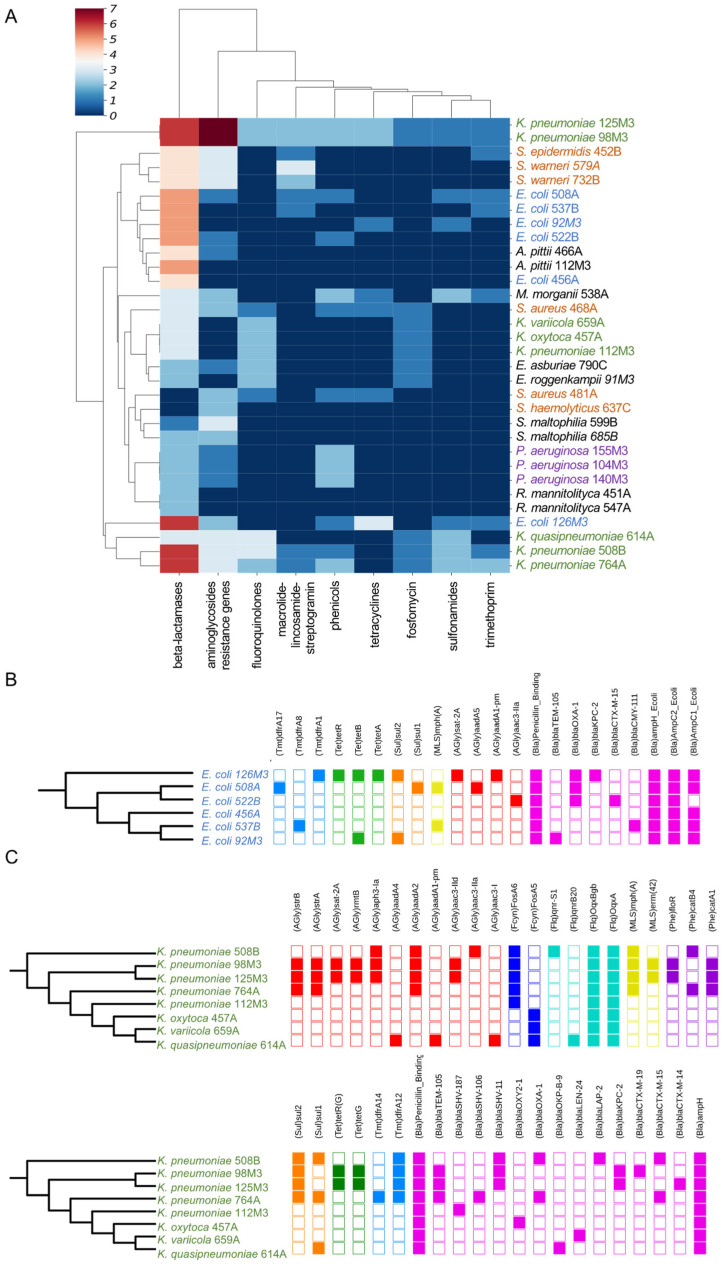
Identification of ARGs in sequenced genomes. (**A**) Heatmap showing the presence of ARGs identified by ARG-ANNOT indicating the number of each resistance gene per genome. Data were clustered using hierarchical mapping with Euclidian distance. The blue to red scale indicates the number of ARGs for each strain in each category, as indicated in the legend. The nucleotide sequences in ARG-ANNOT from different antibiotics classes are abbreviated as follows—AGly: Aminoglycoside resistance genes; Bla: Beta-lactamases; Fcyn: Fosfomycin; Flq: Fluoroquinolones; Gly: Glycopeptides; MLS: Macrolide–lincosamide–streptogramin; Phe: phenicols; Rif: rifampin; Sul: sulfonamides; Tet: tetracyclines; and Tmt: trimethoprim. (**B**) Distribution of different ARGs per genome of *E. coli*, colored by antibiotic category. The maximum likelihood phylogeny for the strains was based on the core genome. (**C**) Distribution of different ARGs per genome of *K. pneumoniae*, following the scheme in (**B**).

**Figure 3 antibiotics-10-00419-f003:**
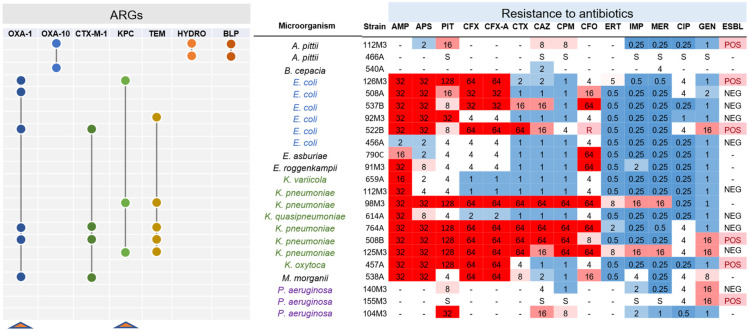
Shared beta-lactamases with 100% identity at the protein level. Seven beta-lactamase coding genes were found as shared among gram-negative strains analyzed. Connected circles indicate that the genes are presented in those strains. On the right, the antibiotic resistance profile of the analyzed strains is shown. Genes for *bla*_OXA-1_ and *bla*_KPC_ are highlighted (red triangles) since these genes were identified in the functional screening carried out in this study. On the right, antibiotic resistance levels are indicated, with numbers indicating the resistance levels in mg/mL. Red indicates resistance to the antibiotic, while blue denotes sensitivity. In the final column it is indicated if the strain is extended-spectrum ß-lactamase (ESBL)-positive or -negative. AMP: ampicillin; APS: ampicillin/sulbactam; PIT: piperacillin/tazobactam; CFX: cefuroxime, CFX-A: cefuroxime axetil; CTX: cefotaxime; CAZ: ceftazidime; CPM: cefepime; CFO: cefoxitin; ERT: ertapenem; IMP: imipenem; MER: meropenem; CIP: ciprofloxacin; GEN: gentamicin.

**Figure 4 antibiotics-10-00419-f004:**
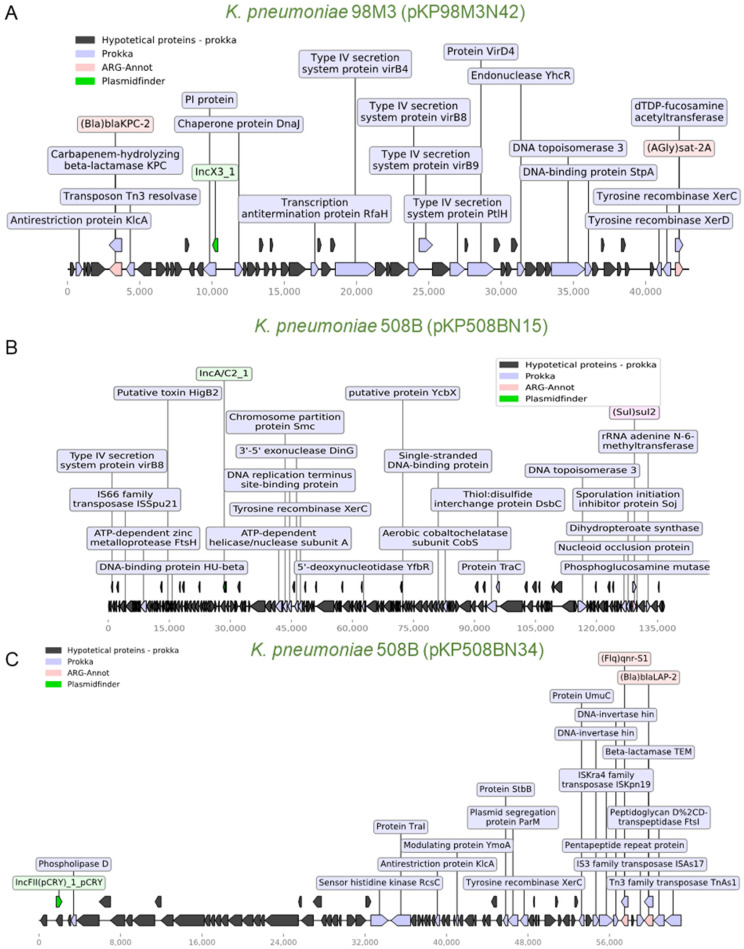
Schematic representation of genes of three *K. pneumoniae* plasmids. (**A**) Plasmid pKP98M3N42 (43 kb) from *K. pneumoniae* 98M3. This plasmid carries two ARGs (*bla*_KPC-2_ and *sat-2A*), elements of a type IV secretion system, two resolvases, and a Tn3 transposase. This plasmid is very similar to pKP125M3N44 from *K. pneumoniae* 125M3 (Appendix A). Whole-plasmid visualization was performed using a python module for prokaryotic genome analyses (DnaFeaturesView) and a matplotlib module combined, as well as ARG-ANNOT and Prokka’s results [49]. (**B**) Plasmid pKP508BN15 (136.8 kb) from *K. pneumoniae* 508B transports *sul2* and an rRNA adenine n-6-methyltransferase (*ermC*) resistance marker. A transposase, a *xerC* recombinase, and a type IV secretion system *virB8* protein are also found in this plasmid. (**C**) Plasmid pKP508BN34 (63 kb) is also from *K. pneumoniae* 508B and carries several transposases and two resistance determinants, *qnr-S1* and *bla*_LAP-2_. Legends represent the colors code for the identified genes.

**Figure 5 antibiotics-10-00419-f005:**
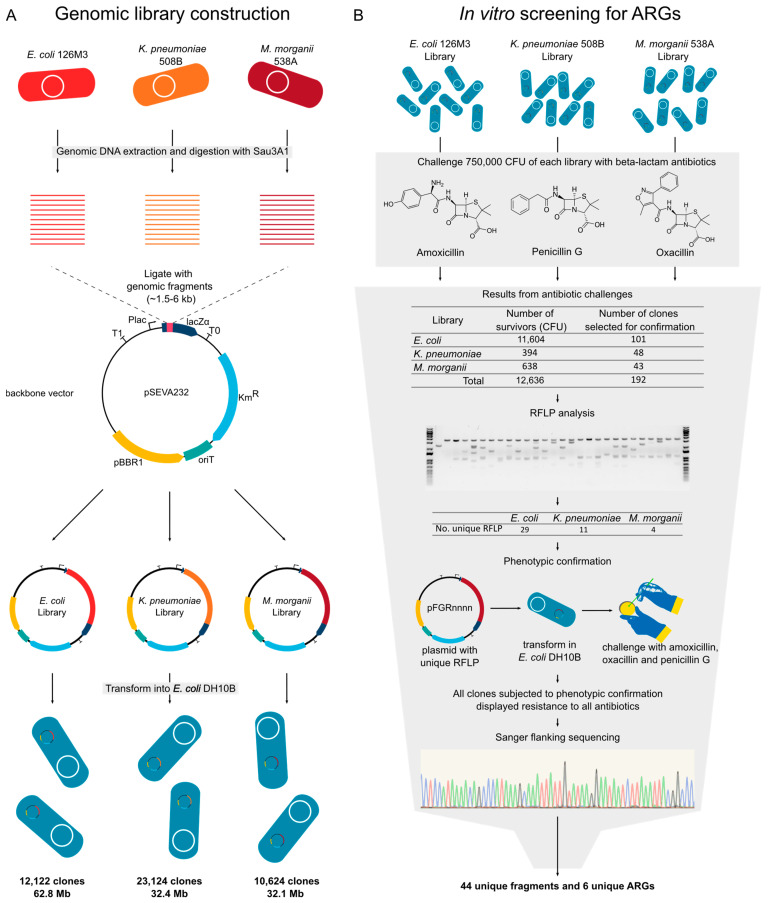
Experimental design for functional genomics analysis. (**A**) Strains and backbone used to construct the library and features of the obtained libraries. (**B**) Functional screening for beta-lactam resistant clones, phenotypic confirmation, and sequence identification. CFU: Colony forming unit; RFLP: Restriction fragment length polymorphism; KmR: Kanamycin resistance marker; pBBR1: A broad host range *oriV*; *lacZα*: *lacZα* gene with multiple cloning site; *Plac*: *Plac* promoter; pFGRnnnn: Plasmid naming schema.

**Figure 6 antibiotics-10-00419-f006:**
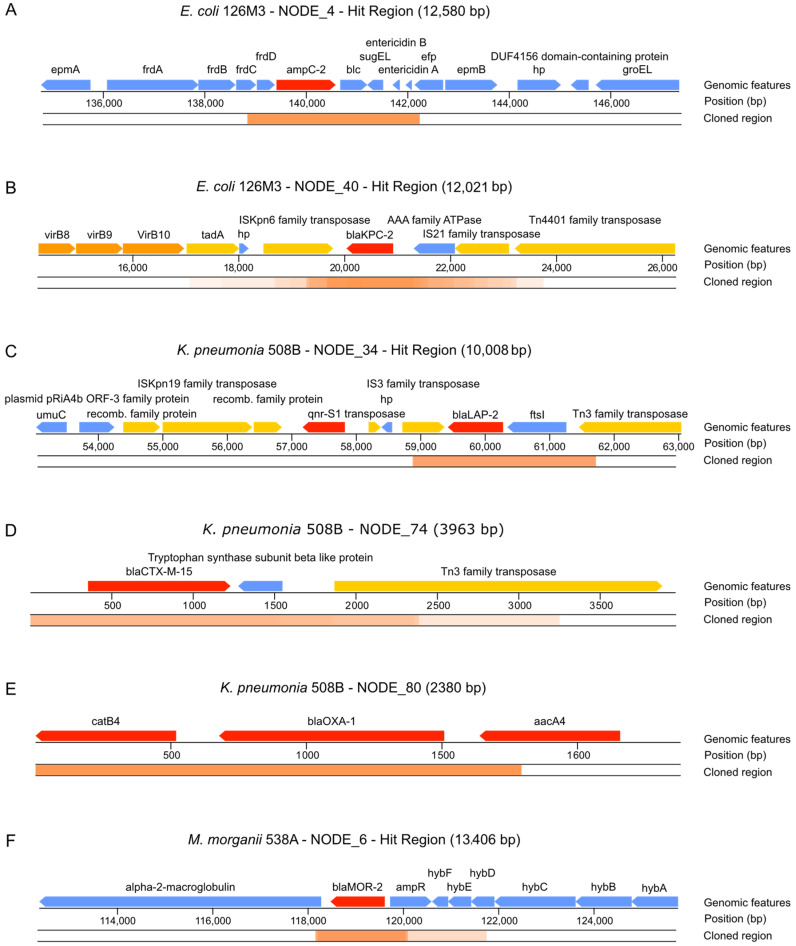
Features found in identified clones conferring resistance to antibiotics. (**A**) *ampC-2* gene identified from *E. coli* 126M3. (**B**) *bla*_KPC-2_ gene identified from *E. coli* 126M3. (**C**) *bla*_LAP-2_ gene identified from pKP508BN34 plasmid from *K. pneumoniae* 508B. (**D**) *blaCTX-M-15* gene identified from *K. pneumoniae* 508B. (**E**) *bla*_OXA-1_ gene identified from *K. pneumoniae* 508B. (**F**) *bla*_MOR-2_ gene identified from *M. morganii* 538A. ORFs are colored according to their function: red—ORFs directly related to antibiotic resistance; orange—ORFs related to virulence; yellow—ORFs related to horizontal transfer of the ARG; blue—ORFs with no identified relation to pathogenesis. The cloned region represents contig regions that were identified in our screenings. Overlapping cloned regions are darker, while dim regions have fewer overlapping identified clones.

**Figure 7 antibiotics-10-00419-f007:**
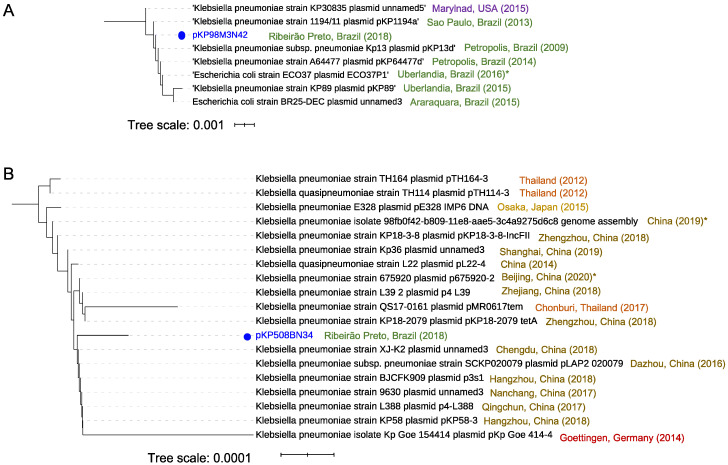
Distance relationship analysis for two plasmids from *K. pneumoniae*. (**A**) Distance relationship of the seven closest plasmids’ nucleotide sequences to pKP98M3N42, showing an *E*-value of less than 0.0 and a minimal sequence cover of 70% in BLAST analysis. The tree was produced using pairwise alignments by means of the fast-minimum evolution method. The year denotes the collection data, and the asterisk indicates the data reported in the public database. (**B**) Distance relationship of the 18 closest plasmids’ nucleotide sequences to pKP508BN34, showing an *E*-value of less than 0.0 and a minimal sequence cover of 50% in BLAST analysis. The tree was produced using pairwise alignments by means of the fast-minimum evolution method. The year indicates the collection data and the asterisk indicates the data reported in the public database, provided when the collection data were not available. iTOL (https://itol.embl.de accessed on 1 December 2020) was used for tree visualization.

**Figure 8 antibiotics-10-00419-f008:**
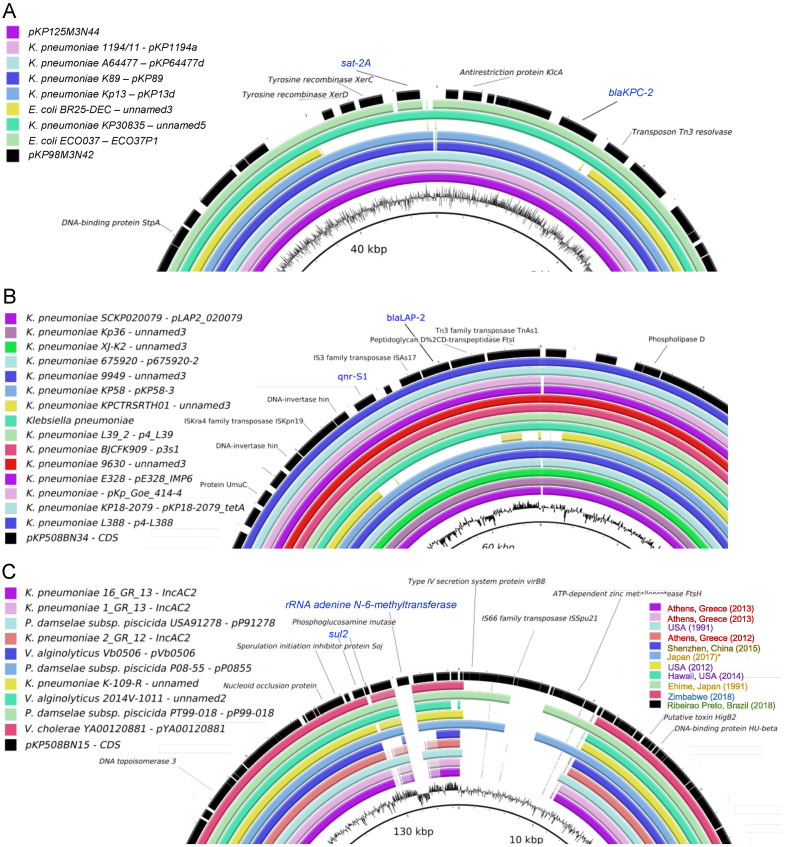
Structural comparison of plasmids from *K. pneumoniae* strains. Identified plasmids were analyzed using blast, and the best hits were used for comparison using BLAST Ring Image Generator (BRIG) [66]. For simplification, only divergent regions between the plasmids are shown. (**A**) pKP98M3N42 (black) and pKP1253N44 (magenta). (**B**) pKP508BN34 (black). (**C**) pKP508BN15 (black).

**Table 1 antibiotics-10-00419-t001:** Features of the generated genomic libraries.

Genomic Library	*E. coli* 126M3	*K. pneumoniae* 508B	*M. morganii* 538A
Total number of clones	13,442	25,693	11,804
Percentage of clones with insert (%)	90	90	90
Number of clones with insert	12,122	23124	10,624
Insert size variation (kb)	1.0–10.5	0.2–2.2	0.2–6.5
Average insert size (kb)	5.2	1.4	3.0
Total genomic library size (mb)	62.8	32.4	32.1
Estimated genome coverage	11.9×	5.8×	7.8×

**Table 2 antibiotics-10-00419-t002:** Comparison of ARG identification using in silico and functional approaches.

Strain	In Silico (Argannot)	Number of Sequenced Resistant Clones
*E. coli* 126M3	Penicillin_Binding_Protein_Ecoli	-
	AmpC1_Ecoli	-
	*bla* _KPC-2_	28
	AmpC2_Ecoli	1
	*bla* _OXA-1_	-
	ampH_Ecoli	-
*K. pneumoniae* 508B	*bla* _SHV-11_	-
	Penicillin_Binding_Protein_Ecoli	-
	*bla* _LAP-2_	2
	ampH	-
	*bla* _CTX-M-15_	8
	*bla* _OXA-1_	1
*M. morganii* 538A	*bla* _CTX-M-15_	-
	*bla* _OXA-1_	-
	*bla* _MOR-2_	4

## Data Availability

All genomes are available at the NCBI under the BioProject number PRJNA641571.

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
