# Peer review of "Combining Functional Genomics and Whole-Genome Sequencing to Detect Antibiotic Resistance Genes in Bacterial Strains Co-Occurring Simultaneously in a Brazilian Hospital"

_antibiotics, 2021, doi:10.3390/antibiotics10040419_

Round 1

Reviewer 1 Report

The present study describes the distribution of antibiotic-resistant genes and mobile-genetic elements among hospital isolates. The authors have sequenced the genome of 34 bacteria and studied the presence of resistance genes and related plasmids within their genome. They have also concluded the genomic study by using Minimum Inhibitory Concentrations (MICs) as a phenotypic test.

However, I think the study lacks a comprehensive analysis regarding the distribution of mobile genetic elements among all sequenced bacteria. Actually, it is a pity that they have the genome of all, but they did not provide information regarding the distribution of phage presence, genomic islands, Integrative and Mobilizable Element (IME), and conjugative elements within their genomes. This contributes to the aim of the study to a great extend. They can apply IslandViwer, ICEberg, and PHASTER free software, respectively, to get information about genomic islands, phage presence, and Mobilizable Elements.

Overall, the strong point about this research is that they used bacteria with different taxonomy instead of focusing on strains. The research has been done comprehensively and correctly. The methodology was properly performed. The discussion is sufficient and covers the results. I think it perfectly fits the Antibiotic’s scope and level. However

Author Response

Reviewer #1 (Comments and Suggestions for Authors):

The present study describes the distribution of antibiotic-resistant genes and mobile-genetic elements among hospital isolates. The authors have sequenced the genome of 34 bacteria and studied the presence of resistance genes and related plasmids within their genome. They have also concluded the genomic study by using Minimum Inhibitory Concentrations (MICs) as a phenotypic test.

However, I think the study lacks a comprehensive analysis regarding the distribution of mobile genetic elements among all sequenced bacteria. Actually, it is a pity that they have the genome of all, but they did not provide information regarding the distribution of phage presence, genomic islands, Integrative and Mobilizable Element (IME), and conjugative elements within their genomes. This contributes to the aim of the study to a great extent. They can apply IslandViwer, ICEberg, and PHASTER free software, respectively, to get information about genomic islands, phage presence, and Mobilizable Elements.

Overall, the strong point about this research is that they used bacteria with different taxonomy instead of focusing on strains. The research has been done comprehensively and correctly. The methodology was properly performed. The discussion is sufficient and covers the results. I think it perfectly fits the Antibiotic’s scope and level.

R: We thank the reviewer for the comments and invaluable suggestions. We used the recommended tools to explore our data and expanded our study with some of those results. We included processed data resulting from our analysis with Phaster for all E. coli and K. pneumoniae strains (focus of the study) in table S2 - supplementary material. Only in one of the genes an intact phage was identified, but it did not overlap with any antibiotic resistance genes.  Data resulting from IslandViewer and ICEberg should be extensively processed before it could be interpreted and included in the current manuscript, which would not be in the scope of our present manuscript, for the sake of rationality we decided to maintain the discussion focused on our findings regarding antibiotic resistance and plasmids.

Reviewer 2 Report

Manuscript ID: antibiotics-1140969 

Review Report

This is a report for the manuscript entitled “Combining functional genomics and whole genome sequencing to detect antibiotic resistance genes in bacterial strains co-occurring simultaneously in a Brazilian hospital”.

This is a reasonably well-written manuscript aiming to demonstrate the relevance of the systematic approach applied by the authors, in the prevention of outbreaks of multidrug resistant bacteria in healthcare facilities.

Apart from the exceptions indicated below in Specific comments, the manuscript is clear, concise, well-documented by clear and adequate figures and tables, also in the Supporting Materials.

Specific comments:

Abstract

Line 33: “…resistant bacteria…”, NOT “…resistance bacteria…”.

Introduction

Line 41: “its” OR: “their”? (twice)

Line 48: “…could even…” OR: “…can even…”

Lines 83 – 94: The content of these five periods is much more an anticipation of the results than the formulation of the objectives of the work.

Results and Discussion

Lines 204 - 207: Please explain this better. Divergent from what? Who are the strains from patients with cystic fibrosis? This information is missing in the legend of Figure S2, which only refers “patients”, while in the figure only one sequence seems to be from a patient with cystic fibrosis.

Lines 300 – 301 (Figure 2 legend): Please remove the sentence about B. cepacia, as it does not correspond to this figure. This sentence apparently refers to Figure 3, where B. cepacia however presents an identified ARG: OXA-10.

Lines 309 – 317 (Figure 3 and legend): The description of abbreviations of the figure is missing, as well as the meaning of ESBL (also referred in the legend).

The first period in line 116 should be removed, as this sentence is repeated in lines 315 – 316, where it makes sense.

Lines 524 – 530 (legend): These sentences should be in the text, not in the legend.

Supporting Materials:

Legend of Figure S2: The referred grey triangles are not shown in this figure. There are repetitions in the legend (in the references to NCBI), so this legend should be re-written.

Author Response

Reviewer #2 (Comments for the Author):

Review Report

This is a report for the manuscript entitled “Combining functional genomics and whole genome sequencing to detect antibiotic resistance genes in bacterial strains co-occurring simultaneously in a Brazilian hospital”.

This is a reasonably well-written manuscript aiming to demonstrate the relevance of the systematic approach applied by the authors, in the prevention of outbreaks of multidrug resistant bacteria in healthcare facilities.

Apart from the exceptions indicated below in Specific comments, the manuscript is clear, concise, well-documented by clear and adequate figures and tables, also in the Supporting Materials.

R: We thank the reviewer for the comments and invaluable suggestions.  We fixed all aspects that were indicated by the reviewer.

Specific comments:

Abstract

Line 33: “…resistant bacteria…”, NOT “…resistance bacteria…”.

R: We fixed this on the manuscript.

Introduction

Line 41: “its” OR: “their”? (twice)

R: We fixed this on the manuscript.

Line 48: “…could even…” OR: “…can even…”

R: We fixed this on the manuscript.

Lines 83 – 94: The content of these five periods is much more an anticipation of the results than the formulation of the objectives of the work.

R: We included two periods (Lines 83-87) summarizing the objectives of the work.

Results and Discussion

Lines 204 - 207: Please explain this better. Divergent from what? Who are the strains from patients with cystic fibrosis? This information is missing in the legend of Figure S2, which only refers “patients”, while in the figure only one sequence seems to be from a patient with cystic fibrosis.

R: We corrected this on the manuscript.

Lines 300 – 301 (Figure 2 legend): Please remove the sentence about B. cepacia, as it does not correspond to this figure. This sentence apparently refers to Figure 3, where B. cepacia however presents an identified ARG: OXA-10.

R: Done.

Lines 309 – 317 (Figure 3 and legend): The description of abbreviations of the figure is missing, as well as the meaning of ESBL (also referred in the legend).

R: We fixed this on the manuscript.

The first period in line 116 should be removed, as this sentence is repeated in lines 315 – 316, where it makes sense.

R: We could not identify what changes were required. Lines were included here for reference.

Line 116

… coli, Pseudomonas aeruginosa, and the genus Staphylococcus. After strain characterization by Vitek 2,…

Lines 315 – 316

… genes were found as shared among gram-negative strains analyzed. Connected circles indicate that the genes are presented in those strains. On the right, the antibiotic resistance profile of the analyzed strains.

Lines 524 – 530 (legend): These sentences should be in the text, not in the legend.

R: We corrected this on the manuscript.

Supporting Materials:

Legend of Figure S2: The referred grey triangles are not shown in this figure. There are repetitions in the legend (in the references to NCBI), so this legend should be re-written.

R: Done.

Reviewer 3 Report

In this study, Borelli et al. investigatd the antibiotic resistance mechanisms in a total of 34 bacterial isolates of several species recovered from an intensive care unit in Brazil. Overall, this is a concise manuscript mainly focusing on the whole genome sequencing data analysis. I just have several concerns which might help the authors to improve their study.

1. It would be better if you can perform the genomic epidemiological analysis and compare the isolates you have sequenced with those deposited in the public database. This information is vital for source tracking bacterial pathogens and also capable to determine the closely related isolates. To perform this analysis, I would like to recommend the authors to use BacWGSTdb 2.0 [Nucleic Acids Research, 49(D1), 2021: D644–D650], a bacterial whole genome sequence typing and source tracking database, which provide a one-stop solution to genomic epidemiological analysis. From this database, you can also catch a glimpse into other isolates belonged to the same sequence type that currently deposited in the NCBI GenBank database. Please also verify if the closest related genomes in the database belong to strains associated with clinical settings or not? If any information regarding isolation source of these strains is available in the database (or even no closely related isolates) then it should be mentioned in the manuscript.

2. Is there any tigecycline and colistin resistance gene in the genome? If so, the MICs of tigecycline and colistin should be reported in the study.

3. Please also report the plasmid replicons in all 34 isolates, which can also be analyzed by BacWGSTdb 2.0.

Author Response

Reviewer #3 (Comments for the Author):

In this study, Borelli et al. investigated the antibiotic resistance mechanisms in a total of 34 bacterial isolates of several species recovered from an intensive care unit in Brazil. Overall, this is a concise manuscript mainly focusing on the whole genome sequencing data analysis. I just have several concerns which might help the authors to improve their study.

  1. It would be better if you can perform the genomic epidemiological analysis and compare the isolates you have sequenced with those deposited in the public database. This information is vital for source tracking bacterial pathogens and also capable to determine the closely related isolates. To perform this analysis, I would like to recommend the authors to use BacWGSTdb 2.0 [Nucleic Acids Research, 49(D1), 2021: D644–D650], a bacterial whole genome sequence typing and source tracking database, which provide a one-stop solution to genomic epidemiological analysis. From this database, you can also catch a glimpse into other isolates belonged to the same sequence type that currently deposited in the NCBI GenBank database. Please also verify if the closest related genomes in the database belong to strains associated with clinical settings or not? If any information regarding isolation source of these strains is available in the database (or even no closely related isolates) then it should be mentioned in the manuscript.

R: We thank the reviewer for the comments and helpful recommendations. Unfortunately, the web-application includes only reference information on K. pneumoniae, E. coli and S. aureus. Therefore, we decided to focus on the genomic epidemiological analysis of K. pneumonia strains 98M and 508B, which were the most relevant for our discussion. We report this information on tables S4 and S5 and it is discussed in the main text.

  1. Is there any tigecycline and colistin resistance gene in the genome? If so, the MICs of tigecycline and colistin should be reported in the study.

R: We did not find any tigecycline or colistin resistance genes in our study.

  1. Please also report the plasmid replicons in all 34 isolates, which can also be analyzed by BacWGSTdb 2.0.

R: We used the recommended tool to explore our data and, due to the absence of reference information on most of our isolates on BacWGSTdb 2.0 we decided to focus on the replicons of K. pneumonia strains 98M and 508B. By employing it we were able to confirm our findings regarding co-occurrence of antibiotic resistance genes and mobilome related sequences. We report this information in table S3 and it is discussed in the main text.

Round 2

Reviewer 3 Report

Thank you for addressing all of my concerns. The revised manuscript is now suitable for publication.